# CONTROLLABLE DIFFUSION VIA OPTIMAL CLASSIFIER GUIDANCE

## ABSTRACT

The controllable generation of diffusion models aims to steer the model to generate samples that optimize some given objective functions. It is desirable for a variety of applications including image generation, molecule generation, and DNA/sequence generation. Reinforcement Learning (RL) based fine-tuning of the base model is a popular approach but it can overfit the reward function while requiring significant resources. We frame controllable generation as a problem of finding a distribution that optimizes a KL-regularized objective function. We present Supervised Learning based Controllable Diffusion (SLCD), which iteratively trains a small classifier to guide the generation of the diffusion model. Via a reduction to no-regret online learning analysis, we show that the output from SLCD provably converges to the optimal solution of the KL-regularized objective. Further, we empirically demonstrate that SLCD can generate high quality samples with nearly the same inference time as the base model in both image generation and biological sequence generation.

## 1 INTRODUCTION

Diffusion models are an expressive class of generative models which are able to model complex data distributions (Song et al., 2021a;b). Recent works have utilized diffusion models for a variety of modalities: images, audio, and molecules (Saharia et al., 2022; Ho et al., 2022; Li et al., 2024; Hoogeboom et al., 2022). However, modeling the distribution of data is often not enough for downstream tasks. We want to generate data which satisfies a specific property, be that a prompt, a specific chemical property, or a specific structure.

Perhaps the simplest approach is classifier-guided diffusion where a classifier is trained using a pre-collected labeled dataset. The score of the classifier is used to guide the diffusion process to generate images that have high likelihood being classified under a given label. However this simple approach requires a given labeled dataset and is not directly applicable to the settings where the goal is to optimize a complicated objective function (we call it reward function hereafter). To optimize reward functions, Reinforcement Learning (RL) and stochastic optimal control based approaches have been studied (Black et al., 2024; Oertell et al., 2024; Domingo-Enrich et al., 2025; Clark et al., 2024; Fan et al., 2023; Uehara et al., 2024a;b). These methods require modifying the base model to some degree which can be slow and expensive to train. We instead turn our attention to "fine-tuning free" methods, which do not modify the base model. Such methods Li et al. (2024) rely on test-time scaling in order walk the Pareto frontier of quality vs divergence from the base distribution. It is thus desirable to devise an algorithm with this same property, but pays a fixed (and small) inference cost.

In this work, we ask the following question: *Can we design a fine-tuning–free algorithm that achieves optimal KL-regularized reward maximization while paying only a fixed, small inference cost?* We provide an affirmative answer to this question (Fig. 1). We view fine-tuning diffusion model as a controllable generation problem where we train a guidance model – typically a lightweight small classifier, to guide the pre-trained diffusion model during the inference time. Specifically, we frame the optimization problem as a KL-regularized reward maximization problem where our goal is to optimize the diffusion model to maximize the given reward while staying close to the pre-trained model. Prior work such as SVDD (Li et al., 2024) also studied a similar setting where they also train a guidance model to guide the pre-trained diffusion model in inference time. However SVDD's solution is sub-optimal, i.e., it does not guarantee to find the optimal solution of the KL-regularized reward maximization objective. We propose a new approach, SLCD (Supervised Learning–based Controllable Diffusion), which **iteratively** refines a classifier using feedback from reward functions

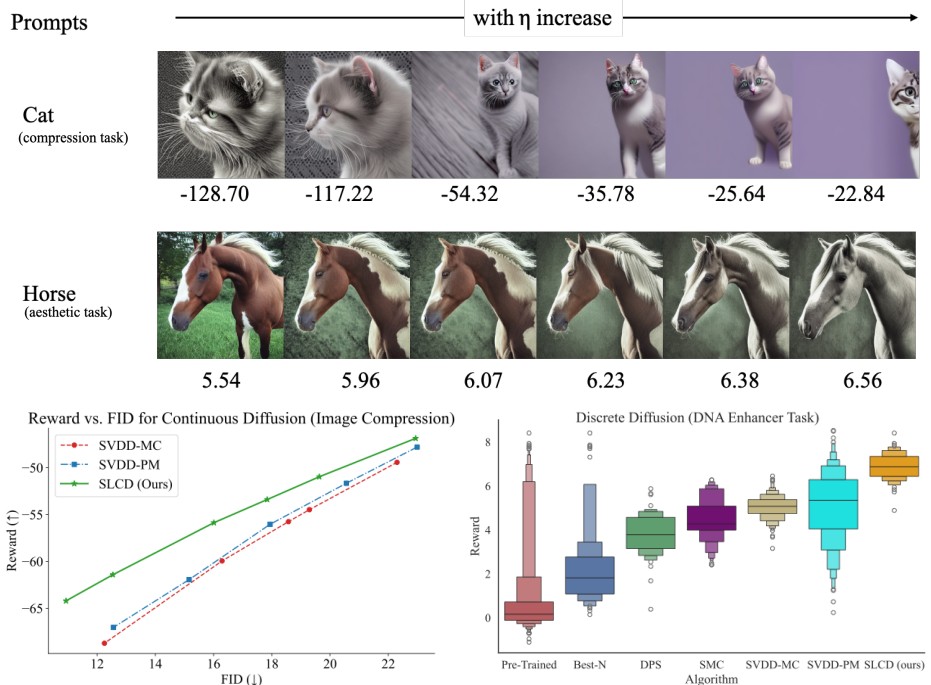

Figure 1: An overview of the main experimental results. **Top:** Qualitative examples for continuous diffusion image tasks (image compression and aesthetic maximization). Relaxing the KL constraint at test time (larger $\eta$) consistently increases the score. **Bottom left:** SLCD stays closer to the initial image distribution (lower FID score) for the same reward. **Bottom right:** SLCD is likewise effective at controlling discrete diffusion models.

applied to the **online data** generated by the classifier itself while guiding a pre-trained diffusion model. Conceptually, SLCD is rooted in classifier guidance, but introduces a novel mechanism for learning the **optimal classifier**–one whose guidance provably ensures that the distribution induced on the base model converges to the desired target distribution.

Through a reduction to no-regret learning (Shalev-Shwartz et al., 2012), SLCD finds a near optimal solution to the KL-regularized reward maximization objective. Our analysis is motivated from the classic imitation learning (IL) algorithms DAgger (Ross et al., 2011) and AggreVaTe(d) (Ross & Bagnell, 2014; Sun et al., 2017) which frame IL as an iterative classification procedure with the main computation primitive being classification. Our theory shows that as long as we can achieve no-regret on the sequence of classification losses constructed during the training, the learned classifier can guide the pre-trained diffusion model to generate a near optimal distribution.

On discrete and continuous diffusion tasks we find SLCD consistently outperforms other baselines on reward and inference speed while maintaining a lower divergence from the base model. Overall, SLCD serves a simple solution to the problem of fine-tuning both continuous and discrete diffusion models to optimize a given KL-regularized reward function.

## 2 RELATED WORK

There has been significant interest in controllable generation of diffusion models, starting from Dhariwal & Nichol (2021) which introduced classifier guidance, to then Ho & Salimans (2022) which introduced classifier-free guidance. These methods paved the way for further interest in controllable generation, in particular when there is an objective function to optimize. First demonstrated by Black et al. (2024); Fan et al. (2023), RL fine-tuning of diffusion models has grown in popularity with works such as Clark et al. (2024); Prabhudesai et al. (2023) which use direct backpropagation to optimize the reward function. However, these methods can lead to mode collapse of the generations and overfitting. Further works then focused on maximizing the KL-constrained optimization problem which regularizes the generation process to the base model (Uehara et al., 2024b) but suffer either from needing special memoryless noise schedulers (Domingo-Enrich et al., 2025) or needing to

control the initial distribution of the diffusion process (Uehara et al., 2024a) to avoid a bias problem. Our approach does not need to do any of these modification. Li et al. (2024) proposed a method to augment the decoding process and avoid training the underlying base model, but yield an increase in compute time. Their practical approach also does not guarantee to learn the optimal distribution. More broadly, Mudgal et al. (2023); Zhou et al. (2025) investigate token-level classifier guidance for KL-regularized controllable generation of large language models. Zhou et al. (2025) also demonstrated that to optimize a KL-regularized RL objective in the context of LLM text generation, one just needs to perform no-regret learning. While our approach is motivated by the Q# approach from Zhou et al. (2025), we tackle score-based guidance for diffusion models in a continuous latent space and continue time, where the space of actions form an infinite set–making algorithm design and analysis substantially more difficult than the setting of discrete token and discrete-time in prior LLM work.

# 3 PRELIMINARIES

## 3.1 DIFFUSION MODELS

Given a data distribution $q_0$ in $\mathbb{R}^d$, the forward process of a diffusion model (Song et al. (2021b)) adds noise to a sample $\bar{\mathbf{x}}_0 \sim q_0$ iteratively, which can be modeled as the solution to a stochastic differential equation (SDE):

$$\mathrm{d}\bar{\mathbf{x}} = \mathbf{h}(\bar{\mathbf{x}}, \tau)\mathrm{d}\tau + g(\tau)\mathrm{d}\bar{\mathbf{w}}, \quad \bar{\mathbf{x}}_0 \sim q_0, \quad \tau \in [0, T] \tag{1}$$

where $\{\bar{\mathbf{w}}_\tau\}_\tau$ is the standard Wiener process, $\mathbf{h}(\cdot, \cdot) : \mathbb{R}^d \times [0, T] \to \mathbb{R}^d$ is the drift coefficient and $g(\cdot) : [0, T] \to \mathbb{R}$ is the diffusion coefficient. We use $q_\tau(\cdot)$ to denote the probability density function of $\bar{\mathbf{x}}_\tau$ generated according to the forward SDE in equation equation 1. We assume $\mathbf{f}$ and $g$ satisfy certain conditions s.t. $q_T$ converges to $\mathcal{N}(0, I)$ as $T \to \infty$. For example, if Eq. (1) is chosen to be Ornstein–Uhlenbeck process, $q_T$ converges to $\mathcal{N}(0, 1)$ exponentially fast.

The forward process equation 1 can be reversed by:

$$\mathrm{d}\mathbf{x} = \left[-\mathbf{h}(\mathbf{x}, T - t) + g^2(T - t)\nabla \log q_{T-t}(\mathbf{x})\right]\mathrm{d}t + g(T - t)\mathrm{d}\mathbf{w}, \quad \mathbf{x}_0 \sim q_T, \quad t \in [0, T], \tag{2}$$

where $\{\mathbf{w}_t\}_t$ is the Wiener process. It is known (Anderson, 1982) that the forward process equation 1 and the reverse process equation 2 have the same marginal distributions. To generate a sample, we can sample $x_0 \sim q_T$, and run the above SDE from $t = 0$ to $t = T$ to get $x_T$. In practice, one can start with $\mathbf{x}_0 \sim \mathcal{N}(0, I)$ (an approximation for $q_T$) and use numerical SDE solver to approximately generate $x_T$, such as the generation processes from DDPM (Ho et al., 2020) or DDIM (Song et al., 2021a).

## 3.2 CONTROLLABLE GENERATION

In certain applications, controllable sample generated from some target conditional distribution is preferable. This can be achieved by adding guidance to the score function. In general, the reverse SDE with guidance $\mathbf{f}(\cdot, \cdot)$ is:

$$\mathrm{d}\mathbf{x} = \left[-\mathbf{h}(\mathbf{x}, T - t) + g^2(T - t)\left(\nabla \log q_{T-t}(\mathbf{x}) + \mathbf{f}(\mathbf{x}, t)\right)\right]\mathrm{d}t + g(T - t)\mathrm{d}\mathbf{w}. \tag{3}$$

For convenience, for all $0 \leq s \leq t \leq T$, we use

$$P^{\mathbf{f}}_{s \to t}(\cdot | p) \tag{4}$$

to denote the marginal distribution of $\mathbf{x}_t$, the solution to equation 3 with initial conditional $\mathbf{x}_s \sim p$. In the remaining of this paper, we may abuse the notation and use $P^{\mathbf{f}}_{s \to t}(\cdot | \mathbf{x}')$ to denote a deterministic initial condition $\Pr[\mathbf{x}_s = \mathbf{x}'] = 1$. In particular, we use $P^{\mathrm{prior}}_{s \to t}(\cdot | p)$ to denote the special case that $\mathbf{f} \equiv 0$.

## 3.3 REWARD GUIDED GENERATION

In this paper, we aim to generate samples that maximize some reward function $r(x) \in [-R_{\max}, 0]$, while not deviating too much from the base or **prior** distribution $q_0$. We consider the setting where we have access to the score function of the prior $q_0$ (e.g., $q_0$ can be modeled by a pre-trained large diffusion model).

Figure 2: Covariate shift (left) and data collection in our approach (right). The left figure illustrates covariate shift. In the offline naive approach, classifier will be trained under the green samples. However in inference time, the classifier will be applied at the red samples – samples generated by using the classifier itself as guidance. The difference in green samples (training distribution) and red samples (testing distribution) is the covariate shift. Our approach (right) mitigates this by iteratively augmenting the training set with samples drawn from guided diffusion. We rollin with the classifier-guided diffusion to get to $\mathbf{x}_t$. We then rollout with the prior's score function to get to $x_T$ and query a reward $r(\mathbf{x}_T)$. The triple $(t, \mathbf{x}_t, r(\mathbf{x}_T))$ will be used to refine the classifier.

Formally, our goal is to find a distribution $p$ that solves the following optimization problem:

$$\max_p \mathbb{E}_{\mathbf{x} \sim p}[r(\mathbf{x})] - \frac{1}{\eta}\mathrm{KL}(p\|q_0) \qquad (5)$$

for some $\eta > 0$ which controls the balance between optimizing reward and staying close to the prior. It is known (Ziebart et al., 2008) that $p^\star$, the optimal solution to the optimization in equation 5, satisfies

$$p^\star(\mathbf{x}) = \frac{1}{Z}q_0(\mathbf{x})\exp(\eta r(\mathbf{x})), \qquad (6)$$

where $Z > 0$ is the normalization factor. Prior work treats this as a KL-regularized RL optimization problem. However as we mentioned, to ensure the optimal solution of the KL-regularized RL formulation to be equal to $p^\star$, one need to additionally optimize the state distribution $q_T$ (e.g., via another diffusion process), or need to design special memory less noise scheduling (Domingo-Enrich et al., 2025). We aim to show that we can learn $p^\star$ in a provably optimal manner.

## 4 ALGORITHM

We introduce a binary label $y \in \{0, 1\}$ and denote the classifier $p(y = 1|\mathbf{x}) := \exp(\eta r(\mathbf{x}))$ (recall that we assume reward is negative). The introduction of the binary label and the classifier allows us to rewrite the target distribution as the posterior distribution given $y = 1$:

$$p(\mathbf{x}|y = 1) \propto q_0(\mathbf{x})p(y = 1|\mathbf{x}) = q_0(\mathbf{x})\exp(\eta r(\mathbf{x})).$$

Given this formulation, the naive approach is to model $p(\mathbf{x}|y = 1)$ via the standard classifier-guided diffusion process. In other words, we generate a labeled dataset $\{(\mathbf{x}, y)\}$ where $\mathbf{x}$ is from the prior $\mathbf{x} \sim q_0$, and the label is sampled from the Bernoulli distribution with mean $\exp(\eta r(\mathbf{x}))$, i.e., $y \sim p(y|\mathbf{x})$. Once we have this data, we can add noise to $\mathbf{x}$, train a time-dependent classifier that predicts the noised sample to its label $y$. Once we train the classifier, we use its gradient to guide the generation process as shown in Eq. (3).

While this naive approach is simple, this approach can fail due to the issue of *covariate shift* – the training distribution (i.e., $q_t$ – the distribution of $\bar{\mathbf{x}}_t$) used for training the classifier is different from the test distribution where the classifier is evaluated during generation (i.e. the distribution of samples $x$ generated during the classifier-guided denoising process). This is demonstrated in left figure in Fig. 2. In the worst case, the density ratio of the test distribution over the training distribution can be exponential $\exp(R_{\max}\eta)$, which can be too large to ignore when $\eta$ is large (i.e., KL regularization is weak).

We propose an iterative approach motivated by DAgger (data aggregation, Ross et al. (2011)) to close the gap between the training distribution and test distribution. First with the binary label $y$ and our definition of $p(y = 1|\mathbf{x}_T) = \exp(\eta r(\mathbf{x}_T))$ (note $\mathbf{x}_T$ represents the generated image), we can show that the classifier $p(y = 1|\mathbf{x}_t)$ for any $t \in [0, T)$ takes the following form:

$$p(y = 1|\mathbf{x}_t) = \mathbb{E}_{x_T \sim P^{\mathrm{prior}}_{t \to T}(\cdot|\mathbf{x}_t)}\exp(\eta r(\mathbf{x}_T)). \qquad (7)$$

---

**Algorithm 1** Controllable diffusion via iterative supervised learning (SLCD)

---

Initialize $\hat{R}^1$.
**for** $n = 1, \ldots, N$ **do**
    set $\mathbf{f}^n$ as Eq. (9).
    collect an additional training dataset $D^n$ following Eq. (10).
    train $\hat{R}^{n+1}$ on $\bigcup_{i=1}^{n} D^i$ according to Eq. (11).
**end for**
**Return** $\mathbf{f}^{\hat{n}}$, the best of $\{\mathbf{f}^1, \ldots, \mathbf{f}^N\}$ on validation.

---

Intuitively $p(y = 1|\mathbf{x}_t)$ models the expected probability of observing label $y = 1$ if we generate $\mathbf{x}_T$ starting from $\mathbf{x}_t$ using the reverse process of the pre-trained diffusion model. We defer the formal derivation to Appendix A which relies on proving that the forward process and backward process of a diffusion model induce the same conditional distributions.

We take advantage of this closed-form solution of the classifier, and propose to model the classifier via a *distributional approach* (Zhou et al., 2025) . Particularly, define $r \sim R^{\text{prior}}(\cdot|\mathbf{x}_t, t)$ as the distribution of the reward of a $\mathbf{x}_T \sim P_{t \to T}^{\text{prior}}(\cdot|\mathbf{x}_t)$. The classifier $p(y = 1|\mathbf{x}_t)$ can be rewritten using the reward distribution $R^{\text{prior}}(\cdot|\mathbf{x}_t, t)$:

$$p(y = 1|\mathbf{x}_t) := \mathbb{E}_{r \sim R^{\text{prior}}(\cdot|\mathbf{x}_t, t)} \exp(\eta \cdot r). \tag{8}$$

Our goal is to learn a reward distribution $\hat{R}$ to approximate $R^{\text{prior}}$ and use $\hat{R}$ to approximate the classifier as $p(y = 1|\mathbf{x}_t) \approx \mathbb{E}_{r \sim \hat{R}(\cdot|\mathbf{x}_t, t)} \exp(\eta r)$. This distributional approach allows us to take advantage of the closed form of the classifier in Eq. (7) (e.g., there is no need to learn the exponential function $\exp(\eta r)$ in the classifier). Algorithm 1 describes an iterative learning approach for training such a distribution $\hat{R}(\cdot|\mathbf{x}_t, t)$ via supervised learning (e.g., maximum likelihood estimation (MLE)).

Inside iteration $n$, given the current reward distribution $\hat{R}^n$, we define the score of the classifier $\mathbf{f}^n$ as:

$$\forall t, x_t : \mathbf{f}^n(\mathbf{x}_t, t) := \nabla_{\mathbf{x}_t} \ln \left( \mathbb{E}_{r \sim \hat{R}^n(\cdot|\mathbf{x}_t, t)} \exp(\eta \cdot r) \right). \tag{9}$$

We use $\mathbf{f}^n$ to guide the prior to generate an additional training dataset $D^n := \{(t, \mathbf{x}_t, r)\}$ of size $M$, where

$$\underbrace{t \sim \text{Uniform}(T), \mathbf{x}_0 \sim \mathcal{N}(0, I), \mathbf{x}_t \sim P_{0 \to t}^{\mathbf{f}^n}(\cdot|\mathbf{x}_0),}_{\textbf{Roll in} \text{ with the score of the latest classifier } \mathbf{f}^n \text{ as guidance}} \quad \underbrace{\mathbf{x}_T \sim P_{t \to T}^{\text{prior}}(\cdot|\mathbf{x}_t), \text{and } r = r(\mathbf{x}_T).}_{\textbf{Roll out} \text{ with the prior to collect reward}} \tag{10}$$

Note that the roll-in process above simulates the inference procedure – $\mathbf{x}_t$ is an intermediate sample we would generate if we had used $\mathbf{f}^n$ to guide the prior in inference. The rollout procedure collects reward signals for $\mathbf{x}_t$ which in turn will be used for refining the reward distribution estimator $\hat{R}(\cdot|\mathbf{x}_t)$. This procedure is illustrated in the right figure in Fig. 2. We then aggregate $D^n$ with all the prior data and re-train the distribution estimator $\hat{R}$ using the aggregate data via supervised learning, i.e., maximum likelihood estimation:

$$\hat{R}^{n+1} \in \arg\max_{R \in \mathcal{R}} \sum_{i=1}^{n} \sum_{(t, \mathbf{x}_t, r) \in D^i} \ln R(r|\mathbf{x}_t, t), \tag{11}$$

where $\mathcal{R}$ is the class of distributions. This rollin-rollout procedure is illustrated in Fig. 2. We iterate the above procedure until we reach a point where $\hat{R}^n(\cdot|\mathbf{x}_t, t)$ is an accurate estimator of the true model $R^{\text{prior}}(\cdot|\mathbf{x}_t, t)$ under distribution induced by the generation process of guiding the prior using $\mathbf{f}^n$ itself. Similar to DAgger's analysis, we will show in our analysis section that a simple no-regret argument implies that we can reach to such a stage where there is no gap between training and testing distribution anymore.

**In the test time**, once we have the score $\mathbf{f}^{\hat{n}}$, we can use it to guide the prior to generate samples via the SDE in Eq. (3). In practice, sampling procedure from DDPM can be used as the numerical solver for the SDE Eq. (3). Another practical benefit of our approach is that the definition of the distribution $R^{\text{prior}}$ and the learned distribution $\hat{R}$ are independent of the guidance strength parameter $\eta$. This means that in practice, once we learned the distribution $\hat{R}$, we can adjust $\eta$ during inference time as shown in Eq. (9) without re-training $\hat{R}$.

**Remark 1** (Modeling the one-dimensional distribution as a classifier)**.** *We emphasize that from a computation perspective, our approach relies on a simple supervised learning oracle to estimate the **one dimensional conditional distribution** $R(r|\mathbf{x}_t, t)$. In our implementation, we use histogram to model this one-dimensional distribution, i.e., we discretize the range of reward $[-R_{\max}, 0]$ into finite number of bins, and use standard **multi-class classification oracle** to learn $\hat{R}$ that maps from $\mathbf{x}_t$ to a distribution over the finite number of labels. Thus, unlike prior work that casts controllable diffusion generation as an RL or stochastic control problem, our approach eliminates the need to talk about or implement RL, and instead entirely relies on standard classification and can be seamlessly integrated with any existing implementation of classifier-guided diffusion.*

**Remark 2** (Comparison to SVDD (Li et al., 2024))**.** *The most related work is SVDD. There are two notifiable differences. First SVDD estimates a sub-optimal classifier, i.e., $\mathbb{E}_{r \sim R^{prior}(\cdot|\mathbf{x}_t, t)}[r]$. The posterior distribution in their case is proportional to $q_0(\mathbf{x}_T) \cdot \mathbb{E}_{r \sim R^{prior}(\cdot|\mathbf{x}_T, T)}[r]$ which is clearly not equal to the target distribution. Second, SVDD does not address the issue of distribution shift and it trains the classifier only via offline data collected from the prior alone.*

We note that our method can also be adapted to discrete diffusion tasks as seen in Section 6. We refer the reader to Appendix E for more details.

## 5 ANALYSIS

In this section, we provide performance guarantee for the sampler returned by Algorithm 1. We use KL divergence of the generated data distribution $P_{0 \to T}^{\mathbf{f}^{\hat{n}}}(\cdot|\mathcal{N}(0, I))$ from the target distribution $p^\star$ to measure its quality. At a high level, the error comes from two sources:

- **starting distribution mismatch:** in the sampling process, we initialize the SDE Eq. (3) with samples from $\mathcal{N}(0, I)$, not the ground-truth $q_T(\cdot|y = 1)$. However, under proper condition, $q_T(\cdot|y = 1)$ converges to $\mathcal{N}(0, I)$ as $T \to \infty$ (see Lemma 3 of Chen et al. (2025)). In particular, when Eq. (3) is chosen to be the OU process, $q_T(\cdot|y = 1)$ converges at an exponential speed: $\text{KL}(\mathcal{N}(0, I) \| q_T(\cdot|y = 1)) = O(e^{-2T})$.

- **estimation error of the guidance:** the estimated guidance $\mathbf{f}^{\hat{n}}$ is different from the ground truth $\nabla_{\mathbf{x}_t} \ln p(y = 1|\mathbf{x}_t)$. But the error is controlled by the regret of the no-regret online learning.

We assume realizability:

**Assumption 3** (realizability)**.** $R^{prior} \in \mathcal{R}$

Our analysis relies on a reduction to no-regret online learning. Particularly, we assume we have no-regret property on the following log-loss. Note $M$ as the side of the each online dataset $D^i$.

**Assumption 4** (No-regret learning)**.** *The sequence of reward distribution $\{\hat{R}^i\}$ satisfies the following inequality:*

$$\frac{1}{NM} \sum_{i=1}^{N} \sum_{(t,\mathbf{x}_t,r) \in D^i} \ln \frac{1}{\hat{R}^i(r|\mathbf{x}_t, t)} - \min_R \frac{1}{NM} \sum_{i=1}^{N} \sum_{(t,\mathbf{x}_t,r) \in D^i} \ln \frac{1}{R(r|\mathbf{x}_t, t)} \le \gamma_N.$$

*where the average regret $\gamma_N = o(N)/N$ shrinks to zero when $N \to \infty$.*

No-regret online learning for the $\log$ is standard in the literature (Cesa-Bianchi & Lugosi, 2006; Foster et al., 2021; Wang et al., 2024; Zhou et al., 2025). Our algorithm implements the specific no-regret algorithm called Follow-the-regularized-leader (FTRL) (Shalev-Shwartz et al., 2012; Suggala & Netrapalli, 2020) where we optimize for $\hat{R}^i$ on the aggregated dataset. Follow-the-Leader type of approach with random perturbation can even achieve no-regret property for non-convex optimization (Suggala & Netrapalli, 2020). This data aggregation step and the reduction to no-regret online learning closely follows DAgger's analysis (Ross et al., 2011).

Under certain condition, the marginal distribution $q_T$ defined by the forward SDE Eq. (1) converges to some Gaussian distribution rapidly (see Lemma 3 of Chen et al. (2025)). For simplicity, we make the following assumption on the convergence:

**Assumption 5** (convergence of the forward process)**.** $KL(\mathcal{N}(0, I) \| q_T(\cdot|y = 1)) \le \epsilon_T.$

For OU processes, $\epsilon_T$ shrinks in the rate of $\exp(-T)$. We assume the reward distribution class satisfies certain regularity conditions, s.t. the estimation error of the classifier controls the score difference:

**Assumption 6.** *There exists $L > 0$, s.t. for all $R, R' \in \mathcal{R}$, and $\mathbf{x}, t$:*

$$\left\| \nabla_{\mathbf{x}} \ln \left( \mathbb{E}_{r \sim R(\cdot|\mathbf{x},t)} \exp(\eta \cdot r) \right) - \nabla_{\mathbf{x}} \ln \left( \mathbb{E}_{r \sim R'(\cdot|\mathbf{x},t)} \exp(\eta \cdot r) \right) \right\|_2$$
$$\leq L \left| \mathbb{E}_{r \sim R(\cdot|\mathbf{x},t)} \exp(\eta \cdot r) - \mathbb{E}_{r \sim R'(\cdot|\mathbf{x},t)} \exp(\eta \cdot r) \right|.$$

Standard diffusion models with classifier guidance train a time-dependent classifier and use the score function of the classifier to control image generation (Song et al., 2021b; Dhariwal & Nichol, 2021). Such assumption is crucial to guarantee the quality of class-conditional sample generation. We defer a more detailed discussion on Assumption 6 to Appendix B. In general such an assumption holds when the functions satisfy certain smoothness conditions.

**Theorem 7.** *Suppose Assumption 3, 4, 5, and 6 hold. There exists $\hat{n} \in \{1, \ldots, N\}$, s.t. $\mathbf{f}^{\hat{n}}$ specified by Algorithm 1 satisfies:*

$$\mathbb{E}\left[ D_{\mathrm{KL}} \left( P_{0 \to T}^{\mathbf{f}^{\hat{n}}}(\cdot | \mathcal{N}(0, I)) \| p^\star \right) \right] \leq \epsilon_T + \frac{1}{2} T \|g\|_\infty^2 L^2 \gamma_N.$$

*where the expectation is with respect to the randomness in the whole training process, and $g$ is the diffusion coefficient defined in Eq. (1).*

Since $P_{0 \to T}^{\mathbf{f}^{\hat{n}}}(\cdot | \mathcal{N}(0, I))$ models the distribution of the generated samples when using $\mathbf{f}^{\hat{n}}$ to guide the prior, the above theorem proves that our sampling distribution is close to the target $p^\star$ under KL. Note that $\epsilon_T$ decays in the rate of $\exp(-T)$ when the forward SDE is an OU process, and $\gamma_N$ decayes in the rate of $1/\sqrt{N}$ for a typical no-regret algorithm such as Follow-the-Learder (Shalev-Shwartz et al., 2012; Suggala & Netrapalli, 2020).

## 6 EXPERIMENTS

We compare SLCD to a variety of training-free and value-guided sampling strategies across four tasks. For Best-of-N, we draw $N$ independent samples from the base diffusion model and keep the one with the highest reward. Diffusion Posterior Sampling (DPS) is a classifier-guidance variant originally for continuous diffusion (Chung et al., 2023), here adapted to discrete diffusion via the state-of-the-art method of Nisonoff et al. (2025). Sequential Monte Carlo (SMC) methods (Del Moral & Doucet, 2014; Wu et al., 2023; Trippe et al., 2022) use importance sampling across whole batches to select the best sample. SVDD-MC (Li et al., 2024) instead evaluates the expected reward of $N$ candidates under an estimated value function, while SVDD-PM uses the true reward for each candidate with slight algorithm modifications. We evaluate on (i) image compression (negative file size) and (ii) image aesthetics (LAION aesthetic score) using Stable Diffusion v1.5 (Rombach et al., 2022), as well as on (iii) 5' untranslated regions optimized for mean ribosome load (Sample et al., 2019; Sahoo et al., 2024) and (iv) DNA enhancer sequences optimized for predicted expression in HepG2 cells via the Enformer model (Avsec et al., 2021). More detailed information regarding the experiments and hyper-parameters used can be found in Appendix E.

In line with Li et al. (2024), we compare the top 10 and 50 quantiles of a batch of generations, in Table 1. We compare to these methods as, like SLCD, all of these methods do not require training of the base model. Overall, we see that SLCD consistently outperforms the baseline methods while requiring nearly the same inference time as the base model, and omitting the need for multiple MC samples during each diffusion step. These four tasks jointly cover two primary application domains of diffusion models: image generation and biological sequence generation, providing a comprehensive assessment of controllable diffusion methods.

### 6.1 REWARD COMPARISON

We compare the reward of SLCD to the baseline methods in Table 1. We see that SLCD is able to consistently achieve higher reward than SVDD-MC and SVDD-PM, and the other baseline methods in all four tasks. The margin of improvement is most pronounced in settings where the classifier closely approximates the true reward, most notably the image compression task, where SLCD nearly attains the optimal reward. To further see the performance of SLCD, we plot the reward distribution

Table 1: Top 10 and 50 quantiles of the generated samples for each algorithm (with 95% confidence intervals). Higher is better. SLCD consistently outperforms the baseline methods. Baseline results taken from Li et al. (2024) as the exact same settings were used.

| Domain | Quantile | Pre-Train | Best-N | DPS | SMC | SVDD-MC | SVDD-PM | **SLCD** |
|---|---|---|---|---|---|---|---|---|
| Image: Compress | 50% | $-101.4_{\pm 0.22}$ | $-71.2_{\pm 0.46}$ | $-60.1_{\pm 0.44}$ | $-59.7_{\pm 0.4}$ | $-54.3_{\pm 0.33}$ | $-51.1_{\pm 0.38}$ | $\mathbf{-13.60_{\pm 0.79}}$ |
| | 10% | $-78.6_{\pm 0.13}$ | $-57.3_{\pm 0.28}$ | $-61.2_{\pm 0.28}$ | $-49.9_{\pm 0.24}$ | $-40.4_{\pm 0.2}$ | $-38.8_{\pm 0.23}$ | $\mathbf{-11.05_{\pm 0.41}}$ |
| Image: Aesthetic | 50% | $5.62_{\pm 0.003}$ | $6.11_{\pm 0.007}$ | $5.61_{\pm 0.009}$ | $6.02_{\pm 0.004}$ | $5.70_{\pm 0.008}$ | $6.14_{\pm 0.007}$ | $\mathbf{6.31_{\pm 0.061}}$ |
| | 10% | $5.98_{\pm 0.002}$ | $6.34_{\pm 0.004}$ | $6.00_{\pm 0.005}$ | $6.28_{\pm 0.003}$ | $6.05_{\pm 0.005}$ | $6.47_{\pm 0.004}$ | $\mathbf{6.59_{\pm 0.077}}$ |
| Enhancers (DNA) | 50% | $0.121_{\pm 0.033}$ | $1.807_{\pm 0.214}$ | $3.782_{\pm 0.299}$ | $4.28_{\pm 0.02}$ | $5.074_{\pm 0.096}$ | $5.353_{\pm 0.231}$ | $\mathbf{7.403_{\pm 0.125}}$ |
| | 10% | $1.396_{\pm 0.020}$ | $3.449_{\pm 0.128}$ | $4.879_{\pm 0.179}$ | $5.95_{\pm 0.01}$ | $5.639_{\pm 0.057}$ | $6.980_{\pm 0.138}$ | $\mathbf{7.885_{\pm 0.231}}$ |
| 5'UTR (RNA) | 50% | $0.406_{\pm 0.028}$ | $0.912_{\pm 0.023}$ | $0.426_{\pm 0.073}$ | $0.76_{\pm 0.02}$ | $1.042_{\pm 0.008}$ | $1.214_{\pm 0.016}$ | $\mathbf{1.313_{\pm 0.024}}$ |
| | 10% | $0.869_{\pm 0.017}$ | $1.064_{\pm 0.014}$ | $0.981_{\pm 0.044}$ | $0.91_{\pm 0.01}$ | $1.117_{\pm 0.005}$ | $1.383_{\pm 0.010}$ | $\mathbf{1.421_{\pm 0.039}}$ |

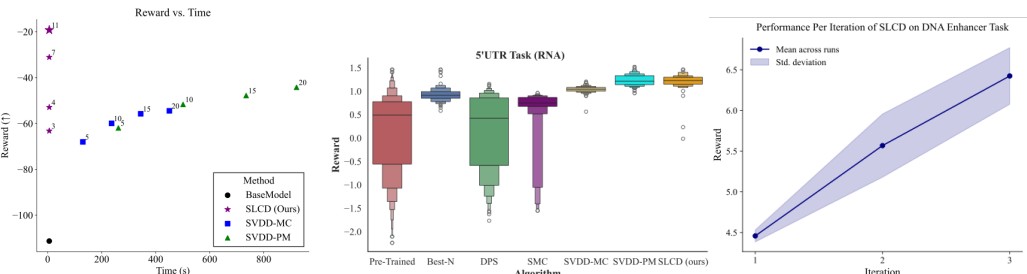

Figure 3: **(Left)** Reward vs. Inference Time on the Compression Task. Numeric labels on SLCD indicate $\eta$, while those on the SVDD denote the duplication number applied at each step.**(Center)** Distribution of rewards for DNA sequences (Enhancers) across different methods. SLCD demonstrates superior performance with higher median and maximum rewards. **(Right)** Reward vs. number of iterations of SLCD. The reward increases as the restart state distribution becomes richer.

of SLCD and the baseline methods in Fig. 3 and Fig. 1. We observe that SLCD produces a more tightly concentrated reward distribution with a higher median reward than the baseline methods, while still maintaining generation diversity, as shown in Fig. 1 with a lower FID score than baseline methods.

## 6.2 QUALITATIVE RESULTS

We present generated images from SLCD in Fig. 4. For the compression task, we observe three recurring patterns: some images shift the subject toward the edges of the frame, others reduce the subject's size, and some simplify the overall scene to reduce file size. For the aesthetic task, the outputs tend to take on a more illustrated appearance, often reflecting a variety of artistic styles.

As $\eta$ increases, the KL constraint is relaxed, enabling a controlled trade-off between optimizing for the reward function and staying close to the base model's distribution. Notably, even under strong reward guidance (i.e., with larger $\eta$), our method consistently maintains a high level of diversity in the generated outputs.

## 6.3 FRÉCHET INCEPTION DISTANCE COMPARISON

Both SLCD and SVDD baselines allow one to control the output sample reward at test time, but via different control variables: SLCD modulates the KL-penalty coefficient $\eta$, while SVDD-MC and SVDD-PM vary the number of Monte Carlo rollouts evaluated at each diffusion step. Since these control parameters can affect sample quality in different ways, we report both the Fréchet Inception Distance (Heusel et al., 2017) (FID) and reward. For the same reward, *higher* FID indicates that the model generates images stray farther from the base models distribution, a sign of reward hacking. We evaluate these methods in Fig. 1. That is, the points on the curve form a pareto frontier between reward and FID. SLCD is able to achieve a better reward-FID trade-off than SVDD-MC and SVDD-PM.

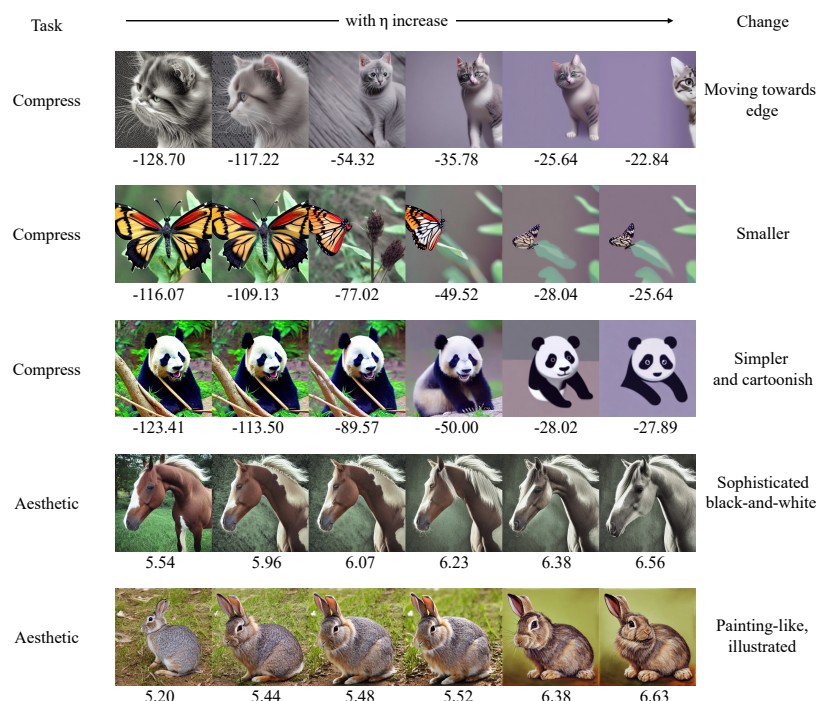

Figure 4: Images generated by SLCD with varying $\eta$ values and their rewards . The first column shows results from the base model, SD1.5, which corresponds to our method with $\eta = 0$. As $\eta$ increases, the KL penalty is relaxed, allowing the generated images to be more strongly optimized for the reward function, and consequently, they diverge further from the base model's original distribution.

## 6.4 INFERENCE TIME COMPARISON

An additional advantage of SLCD is its negligible inference overhead at test time, even when using higher $\eta$ values to achieve greater rewards. In Fig. 3, we compare the wall-clock generation time per image on an NVIDIA A6000 GPU for SLCD against SVDD-MC and SVDD-PM. SLCD achieves higher rewards while requiring significantly fewer computational resources and substantially shorter inference times than either baseline. Specifically, SLCD takes only 6.06 seconds per image, nearly identical to the base model SD 1.5's 5.99 seconds.

Importantly, unlike SVDD methods that incur increased computational cost to improve rewards, SLCD maintains constant inference time across all $\eta$ values, achieving enhanced performance with no additional computation.

## 6.5 ABLATION STUDY

To elucidate the impact of each training cycle, we vary the number of SLCD iterations and plot the resulting reward in Fig. 3. As additional iterations enrich the state distribution and mitigate covariate shift for the classifier, the reward consistently rises. This confirms that our iterative approach can mitigate covariate shift issue. In practice, we only require a small number of iterations to achieve high reward (for example, 3 for the compression task)

Because the scaling parameter $\eta$ can be chosen at test time when the distribution $\hat{R}$ is fully trained, SLCD enables test-time control over the KL penalty during inference. By modulating $\eta$ at test-time, practitioners can smoothly trade-off reward against sample quality without retraining the distribution $\hat{R}$, as demonstrated in Fig. 4.

## 7 CONCLUSION

In this work, we introduced SLCD, a novel and efficient method that recasts the KL-constrained optimization problem as a supervised learning task. We provided theoretical guarantees showing that SLCD converges to the optimal KL-constrained solution and how data-aggregation effectively

mitigates covariate shift. Empirical evaluations confirm that SLCD surpasses existing approaches, all while preserving high fidelity to the base model's outputs and maintaining nearly the same inference time.

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
