# OpenReview forum: "Controllable Diffusion via Optimal Classifier Guidance"
_ICLR.cc/2026/Conference — Submitted to ICLR 2026_

### Official Review · Reviewer_9m6m · 2025-10-30

**Soundness:** 2
**Presentation:** 3
**Contribution:** 2
**Rating:** 4
**Confidence:** 5

**Summary:**

The paper proposes Supervised Learning–based Controllable Diffusion, a KL-regularized reward maximization as optimal classifier guidance for a pre-trained diffusion model. SLCD trains a small time-dependent classifier by iteratively rolling in with the current guidance and rolling out with the prior to collect rewards, and uses a distributional estimator of the reward at each diffusion time to compute the optimal guidance score, which also enables test-time adjustment of the KL penalty. Theoretical analysis reduces learning to no-regret online learning. Empirically, the paper reports reward and trade-offs on image, 5’ UTR, and enhancer tasks.

**Strengths:**

1. The reduction to no-regret learning and the KL bound on the sampling distribution is an interesting theoretical perspective.

2. The roll-in/roll-out data aggregation to align train/test distributions for the classifier is motivated.

3. Finetuning-free with small inference overhead is a appealing goal to achieve.

**Weaknesses:**

1. Missing RL fine-tuning baselines are needed, such as DPOK and Direct reward backprop on diffusion, which are RL/fine-tuning approaches to reward-aligned diffusion. These represent the “train-the-model” side of the Pareto frontier the paper positions against, and without them the empirical claim of superiority over existing approaches is incomplete.
In addition, discrete-guidance baseline [1] is missing, which is a guidance framework for discrete state-space DMs directly overlapping with your DNA/RNA settings.

2. DPS configuration is not fully specified and can be potentially unfavorable. The paper should examine stronger DPS variants, since SLCD is rooted in classifier guidance.

3. The learning process requires substantial computation compared to the inference, which is not well-analyzed or quantified. This also makes the claim of "paying only a fixed, small inference cost" less convincing.

4. Assumptions in theory are strong and partially opaque.
The realizability (Assumption. 3), no-regret (Assumption. 4) and smoothness bound linking distributional error to score error (Assumption. 6) are non-trivial and central to Theorem 7. The paper would benefit from concrete instantiations (e.g., Lipschitz constants) and a finite-sample bound on N,M required for optimality.

5. Ablations on iteration count are light. Few ablation studies are provided to show the contributions of the key components. Fig. 3 shows reward improves with iterations, but doesn’t analyze variance, overfitting, or the trade-off vs. online dataset size. Also, the paper discretize rewards and learn a multi-class classifier, but there’s no ablation on binning strategy, calibration, or on alternative parametric families.

6. Fig. 4 notes images “move subjects to edges,” “simplify scenes” for compression. This looks like reward hacking; the paper should quantify semantic drift vs. reward gain (e.g., CLIP-sim, caption fidelity), not only FID, to support “stays close to base distribution” claims.

7. Code is not provided, and thus readers cannot verify exact reproducibility.

[1] Unlocking Guidance for Discrete State-Space Diffusion and Flow Models

**Questions:**

Could stronger DPS settings close the gap between classifier guidance baseline and the proposed method?

---

> ### Author Response · Authors · 2025-11-24
>
> We thank the reviewer for their review.
>
> ### Response to Weaknesses
>
> > **Missing RL fine-tuning baselines are needed, such as DPOK and Direct reward backprop on diffusion.**
>
> 1\. SLCD allows flexibility of choosing different $\\eta$ to balance between reward maximization and staying close to the prior. But RL does not allow this flexibility. Empirically, we compare the reward and FID score of SLCD, DDPO \[4\], and SVDD. To study the reward-KL tradeoff for DDPO, we evaluate the reward and FID of DDPO for selected intermediate checkpoints. We observe that SLCD gets a higher reward for a fixed FID and a smaller FID for a fixed reward. SLCD dominates the pareto frontier here for all tested classes of FID and Reward tradeoffs. Please see appendix G of the updated PDF for this comparison.
>
> 2\. SLCD has the same convergence rate as online supervised learning, which is much faster than the convergence rate of RL. Empirically, we compare SLCD with DDPO \[4\]. In the case of compression, we observe that DDPO achieves less than \-30 reward after 10k reward queries., while SLCD is able to achieve maximal reward of \-13 after only 8,400 reward queries.
>
> 3\. SLCD only requires training a light weight classifier while RL requires training the whole model, allowing training with limited compute. For example, for the aesthetic task, SLCD only needs to train a 10MB classifier while RL needs to fine-tune the 4GB model.
>
> Finally, we note that we do indeed use \[3\] (cited by the reviewer) as our comparison baseline in the discrete diffusion case. Likewise, we use \[1\] for image generation tasks.
>
> > **DPS configuration is not fully specified and can be potentially unfavorable. The paper should examine stronger DPS variants, since SLCD is rooted in classifier guidance.**
>
> We used the same settings as SVDD \[2\], using \[1\] and \[3\] for continuous and discrete diffusion.
>
> > **The learning process requires substantial computation compared to the inference, which is not well-analyzed or quantified. This also makes the claim of "paying only a fixed, small inference cost" less convincing.**
>
> First, we first point the reviewer to our response to uwJ5 in which we discuss that SLCD is more sample efficient than RL-based methods.
>
> However, when considering methods that require less compute, we believe that this extra compute cost is marginal compared to the time required to train large models. Further, to generate \~ \-45 compressibility reward, it would take around 15 minutes for generation of SCLD-PM, compared to 30 seconds for SLCD (see figure 3 left) in the case of compressibility. Training the compressibility classifier took only 5hr 20min on a single a6000 gpu.
>
> > **Assumptions in theory are strong and partially opaque. The realizability (Assumption. 3), no-regret (Assumption. 4\) and smoothness bound linking distributional error to score error (Assumption. 6\) are non-trivial and central to Theorem 7\. The paper would benefit from concrete instantiations (e.g., Lipschitz constants) and a finite-sample bound on N,M required for optimality.**
>
> The realizability assumption and no-regret assumption are standard in the online learning literature (see e.g. DAgger by Ross et al. 2011 and  Foster et al. 2021\) Please refer to our line 314 to 319 for the detailed discussion. In the paper, we hope to emphasize our algorithmic design and underlying principles rather than illustrate the no-regret algorithm itself.
>
> Please refer to Appendix B regarding Assumption 6, where we discuss and rigorously prove when this assumption may hold. At a high level, Assumption 6 only relies on some smoothness conditions of the problem instance.
>
> > **Ablations on iteration count are light. Few ablation studies are provided to show the contributions of the key components. Fig. 3 shows reward improves with iterations, but doesn’t analyze variance, overfitting, or the trade-off vs. online dataset size. Also, the paper discretizes rewards and learns a multi-class classifier, but there’s no ablation on binning strategy, calibration, or on alternative parametric families.**
>
> The two main differences that we introduce with respect to the algorithm are (1) the iterative learning structure and (2) the contribution of using binning combined with connecting this to KL regularization. Although there are many variables we can ablate, we mainly focused on mean reward and indeed showed improvement in Fig. 3\. With respect to binning, we found on the compressibility task that large (1k) bins significantly degraded the quality of the reward (around \-65.2 reward), while (200 vs 100\) bins had marginal difference (-30.2 vs \-36.5). Using 5 bins had similar results to our 50 bin choice for the experiments (-16.1). We hypothesize that increasing the number of bins to be too large means that each bucket does not have enough data and thus provides an inaccurate estimate. This ablation is now added to the appendix.

---

> > ### Author Response · Authors · 2025-11-24
> >
> > > **Fig. 4 notes images “move subjects to edges,” “simplify scenes” for compression. This looks like reward hacking; the paper should quantify semantic drift vs. reward gain (e.g., CLIP-sim, caption fidelity), not only FID, to support “stays close to base distribution” claims.**
> >
> > Thank you for the suggestion. While additional metrics such as CLIP similarity or caption fidelity can certainly provide further insight into semantic drift, in this work we primarily use **FID** because it serves as a practical *proxy* for distributional closeness—precisely the quantity that our objective aims to control. Our training objective is designed to remain close to the **base distribution in a KL sense**, and FID, although imperfect, is a widely adopted approximation of this type of distributional divergence. Importantly, our objective does not constrain *how* closeness is maintained (e.g., along semantic or structural dimensions), but rather enforces closeness at the distributional level.
> >
> > That said, we agree that measuring semantic drift using metrics like CLIP-similarity or caption fidelity would provide a richer picture of how SLCD balances reward improvement and distribution preservation. We consider this an interesting direction for deeper analysis and plan to explore it in future work.
> >
> > > **Code is not provided, and thus readers cannot verify exact reproducibility.**
> >
> > We have uploaded the code to the supplemental material to allow ease of reproducibility.
> >
> > ### Response to Questions
> >
> > > **Could stronger DPS settings close the gap between classifier guidance baseline and the proposed method?**
> >
> > We do not believe that other DPS settings could close the gap. This is as we used the same settings as SVDD \[2\], using \[1\] and \[3\] for continuous and discrete diffusion. For the discrete diffusion case, we believe that \[3\] is indeed the best performing DPS to our knowledge.
> >
> > \[1\] Chung, H., J. Kim, M. T. Mccann, M. L. Klasky, and J. C. Ye (2022). Diffusion posterior sampling
> > for general noisy inverse problems. arXiv preprint arXiv:2209.14687.
> > \[2\] X Li, et al. Derivative-Free Guidance in Continuous and Discrete Diffusion Models with Soft Value-Based Decoding
> > \[3\] Nisonoff, H., J. Xiong, S. Allenspach, and J. Listgarten (2024). Unlocking guidance for discrete
> > state-space diffusion and flow models. arXiv preprint arXiv:2406.01572
> > \[4\] K Black, et al. Training Diffusion Models with Reinforcement Learning

---

### Official Review · Reviewer_uwJ5 · 2025-11-01

**Soundness:** 2
**Presentation:** 3
**Contribution:** 2
**Rating:** 2
**Confidence:** 3

**Summary:**

This paper introduces Supervised Learning based Controllable Diffusion (SLCD), a new method for steering pre-trained diffusion models to generate samples that optimize a given reward function. The authors frame this as a KL-regularized optimization problem. The core idea is to iteratively train a lightweight classifier to guide the diffusion process. To address the covariate shift inherent in training such a classifier on offline data, SLCD employs an iterative data aggregation (DAgger-like) strategy, where the classifier is repeatedly retrained on new data generated by the guided model itself. The paper provides a theoretical analysis showing that this approach, via a reduction to no-regret online learning, converges to the optimal solution of the KL-regularized objective. Empirical results on image generation and biological sequence tasks show that SLCD can generate high-quality samples that achieve high rewards, maintains a good reward-FID trade-off, and adds negligible computational overhead at *inference* time.

**Strengths:**

The paper is well-written, and the core idea is presented clearly and intuitively. The formulation of controllable generation as a KL-regularized objective solved by an iterative classifier is easy to follow.

The approach of using a DAgger-style iterative training loop to mitigate covariate shift for a guidance classifier appears to be a novel contribution to this problem space. To my knowledge, this specific application is new.

The theoretical analysis that reduces the problem to no-regret online learning provides a solid foundation for the algorithm's convergence, which is a significant strength.

**Weaknesses:**

A primary concern is the fairness of the experimental comparison in Table 1. The paper compares SLCD against methods like Best-N, DPS, SMC, and SVDD. It is noted that these baselines are "training-free" or can use an off-the-shelf classifier (or value function) trained a single time on offline data. However, SLCD introduces a significant computational overhead through its *iterative* training and data generation loop (Algorithm 1). This iterative refinement is a form of training that the baselines do not have. The direct comparison of final sample quality (reward) may not be fair, as SLCD benefits from a much more intensive procedure to create its guidance model. The paper emphasizes its low *inference* cost but does not discuss the *training* cost of the guidance model itself, which is not required by most of the chosen baselines.

The experiments are missing a crucial baseline comparison. The paper motivates its approach by highlighting the covariate shift issue of standard classifier guidance (where a time-dependent classifier is trained offline). However, this standard classifier guidance method is never empirically compared against. Without this baseline, it is difficult to quantify how much of the performance gain is due to the novel iterative training scheme versus simply using any form of classifier guidance (even a "naive" offline one).

I currently rate this paper as reject due to this issue. If I missunderstand something, please clarify and I am happy to adjust the score if this issue is addressed.

**Questions:**

1. Algorithm 1 mentions a "validation" step (Line 224) to select the best classifier, $f^{\hat{n}}$, from the $N$ iterations. What is the detailed process for this validation? How is the validation set constructed, and what metric is used to determine the "best" model?
2. Related to the weakness section, could the authors provide clarification on the total computational cost (e.g., training time or GPU-hours for the *guidance model*) for SLCD? This would be necessary to fairly assess the trade-offs against methods like DPS, which only require a one-time offline training of their value function (usually considered as "training-free guidance" in literature).
3. Why was standard, time-dependent classifier guidance (trained offline on the prior distribution) omitted as a baseline in the experiments? Including this comparison would seem essential to empirically validate the paper's core claim that the iterative, on-policy data collection is necessary to overcome covariate shift.

---

> ### Author Response · Authors · 2025-11-24
>
> We thank the reviewer for their review.
>
> #### Response to Weaknesses
>
> > **SLCD requires more training compared to other baselines**
>
> Thank you for raising this concern. We agree that SLCD involves an up-front training stage, whereas several baselines (e.g., Best-of-N, DPS, or SMC) can operate using an off-the-shelf classifier. However, we believe the comparison remains fair for three reasons.
>
> **(1) Baselines incur substantially higher inference cost.**
>
> Methods such as SVDD-MC, SVDD-PM, and SMC require repeated model evaluations or multi-step sampling at inference time. This cost is incurred *every time* generation occurs, and it scales linearly with the amount of data generated. In contrast, SLCD pays a fixed up-front cost to train a lightweight classifier, after which generation is extremely efficient: sampling uses only a single forward pass per diffusion step, identical to standard classifier guidance. For real applications where many generations are required, amortizing this cost makes SLCD significantly more practical.
>
> **(2) The classifier training cost is modest.**
>
> Although SLCD uses an iterative refinement loop, the classifier itself is small and fast to train. For example, in the compressibility task, the full SLCD pipeline takes roughly **5 hours on a single A6000 GPU**. This is a small fixed cost compared to the repeated heavy inference performed by the baselines.
>
> **(3) SLCD is highly sample-efficient compared to RL-based fine-tuning.**
>
> Prior fine-tuning methods that rely on RL fine-tuning require large numbers of model rollouts and expensive optimization (e.g., PPO-style updates). SLCD avoids RL entirely: each refinement round only requires supervised learning on classifier-labeled samples, making it much faster than RL-based approaches. For example, in the case of compression, we observe that DDPO achieves less than \-30 reward after 10k reward queries., while SLCD is able to achieve maximal reward of \-13 after only 8,400 reward queries.
>
> Overall, although SLCD introduces a modest up-front training step, it yields far cheaper inference, strong controllability, and much better sample efficiency than RL-based fine-tuning.
>
> > **Missing naive baseline.**
>
> We point the reviewer to Figure 3, right, in which we plot iteration vs reward. We note that the first iteration consists of training the classifier on a fixed dataset generated from the unguided diffusion model. The plot shows a strong increase in reward as the number of iterations increases. Likewise, this pattern is shown in other tasks: in the DNA task, after initial reward was approximately 4.5 (vs 7.4 with SLCD), while for the RNA task was 0.7 (vs 1.31 from SLCD). Likewise, for the compression task was \-50.961 (vs \-13.6 from SLCD)  and for the aesthetic task was 5.89 (vs. 6.3 from SLCD)
>
> #### Response to Questions
>
> > **How is the classifier chosen?**
>
> Choosing the best classifier on validation is for theoretical considerations, a common approach in the reinforcement learning literature. See e.g. Alg 3.1 of DAgger (Ross et al. 2011). In the tasks tested, we selected the classifier which achieved the highest reward for a fixed $\\eta$. However, in the case that reward is hard to access, using explained variance showed high correlation with classifier quality in our experiments.
>
> > **Could the authors provide clarification on the total computational cost (e.g., training time or GPU-hours for the *guidance model*)?**
>
> The time for each task varies significantly, but for example, both the compression and DNA task take a little over 5 GPU hours on an A6000, to generate images and train the classifier. The RNA task took 8 GPU hours to complete. The generation and training tasks are highly parallelisable, which depending on resources can reduce wall clock time.

---

> > ### Comment · Reviewer_uwJ5 · 2025-11-26
> >
> > Thank you for the detailed response and clarifications. I’m mostly satisfied regarding the “naive” classifier baseline (Fig. 3), especially if you clarify in the paper that the first iteration corresponds to a classifier trained on a fixed offline dataset from the unguided model.
> >
> > My remaining concerns are mainly about computational cost and clarity of comparison:
> >
> > 1. While I agree that some TFG methods (e.g., SVDD, SMC) have heavy inference, others such as DPS and methods like MPGD [1] are much lighter. Under the “training-free guidance” umbrella, it would be helpful to more carefully compare computational overhead against a broader set of TFG techniques, including [1,2]. Also, 5+ GPU hours of training can be larger than simply running these TFG methods on the same sample budget, especially for small or moderate-scale generation. If one were allowed to collect samples from TFG methods and then train a classifier, the amortized-cost argument becomes less clear. Fig. 3 is also hard to interpret in terms of cost, since training time per iteration is not reflected.
> > 2. Please include the validation procedure in the paper. For the claim that explained variance correlates with classifier quality, some quantitative evidence is necessary instead of the vague description "high correlation".
> > 3. The statements “a little over 5 GPU hours” or “8 GPU hours” are too vague. Since efficiency is a key claim, I encourage you to report precise GPU-hours for each task, broken down into (i) data generation, (ii) classifier training, and (iii) any extra evaluation, and to compare these numbers to baselines such as DPS/MPGD/TFG where possible.
> >
> > I am fine with the baseline story in Fig. 3 once these clarifications are incorporated.
> >
> > [1] He et al., *Manifold Preserving Guided Diffusion*, ICLR 2024.
> >  [2] Ye et al., *TFG: Unified Training-Free Guidance for Diffusion Models*, NeurIPS 2024.

---

> > > ### Author Response · Authors · 2025-12-04
> > >
> > > We thank the reviewer for their comments.
> > >
> > > > **MPGD should be compared to SLCD.**
> > >
> > > From a theoretical perspective, MPGD relies on a **biased estimate** approximating $r(E\[x\_0 | x\_t\])$ as $E\_{x\_0 | x\_t} \[r(x\_0)\]$, which is not equal when $r$ is nonlinear. SCLD however relies on an **unbiased approach** to gradient estimation. As mentioned in the TFG paper, the theoretical difference between gradients reliant on the clean image approximation and the noised image is unclear, in contrast to the gradient in SLCD's case. Furthermore, this method is unable to vary at test-time the KL constraint, a feature of design in SLCD.
> > >
> > > Finally, we note that SLCD outperforms DPS, a strong baseline for training free guidance methods.
> > >
> > > > **Validation procedure was too vague and needs to be included in the paper**
> > >
> > > The use of explained variance is for determining the *classifier quality* (ie, in addition to loss, what other metrics can be taken into consideration to see if the classifier is able to learn the distribution) not selecting the classifier during training. When determining which checkpoint to select, we take the final checkpoint as reward increases during each iteration of training. As mentioned previously, the “best on validation” is for theoretical purposes.
> > >
> > > > **Training time breakdown.**
> > >
> > > We report the exact training times as follows: the compressibility task required **5 hours 18 minutes**, the RNA task **8 hours 13 minutes**, and the DNA task **4 hours 53 minutes**. Please refer to Figure 3 (left) for an inference-time comparison with the baseline method. Since training-based and training-free methods have fundamentally different training and inference computational costs, achieving a strictly fair comparison is not straightforward.

---

### Official Review · Reviewer_cy8t · 2025-11-01

**Soundness:** 3
**Presentation:** 4
**Contribution:** 3
**Rating:** 4
**Confidence:** 3

**Summary:**

This paper proposes Supervised Learning based Controllable Diffusion (SLCD), a classifier-guided diffusion method where the guidance signal is provided by a classifier trained to estimate step-wise reward values along the diffusion trajectory. A key challenge of classifier guidance is covariate shift between real-data distributions used for classifier training and intermediate latent distributions encountered during generation (inference). SLCD addresses this by training the classifier on samples drawn from intermediate diffusion steps, thereby aligning the classifier’s training distribution with its test-time usage. Experiments on image generation tasks and a DNA enhancement tasks demonstrates that SLCD yields improved controllability with negligible increase in inference cost.

**Strengths:**

- The method is supported by clear theoretical motivation and formal analysis explaining how SLCD reduces covariate shift compraed to prior classifier-guided approaches.
- The visualization and derivation in Figure 2 effectively illustrate the benefit of aligning classifier training distribution and inference distribution.
- The framework is conceptually simple, inference-time effective, compatible with existing diffusion models, and does not require modifying the base model architecture.

**Weaknesses:**

- Important considerations—including classifier architecture, hyperparameters, sampling strategy for intermediate steps, and reward definitions—are only provided in the supplementary material without directions in the main script. Some guidance should be included in the main paper.
- While classifier training is central to SLCD, the paper does not analyze how classifier quality influences controllability or generation performance (e.g., ablation over classifier capacity, generalization on prompts, or data size).
- Experimental validation on image tasks is limited to SD 1.5, which is no longer state-of-the-art. Evaluation on stronger models (e.g., SDXL or SD3) would strengthen the claim that SLCD generalizes broadly.
- The scaling parameter for classifier guidance plays a crucial role, but no discussion is provided regarding sensitivity or tuning strategy.

**Questions:**

- Does Equation (5) compute KL divergence over the conditional distributions or marginal distributions? If samples are collected under conditional generation in SLCD, is q_0 treated as a conditional distribution as well?
- Could the authors clarify the motivation for using the compression task as an image evaluation setting? What intuition links compression to controllable guidance?
    - Also, could the authors provide more details about the compression task?
- How are hyperparameters for classifier training choosen? What is the computational cost (both sample collection and classifier training)? Does the classifier generalize to unseen prompts?
- How is the scaling parameter chosen in practice? Is a single value applied across experiments? (such as different rows in Figure 4), or is it tuned per dataset/task?
- In Section 6.3, is FID computed between base SD outputs and SLCD outputs, or between generated outputs and real images?

---

> ### Author Response · Authors · 2025-11-24
>
> We thank the reviewer for their review.
>
> #### Response to Weaknesses
>
> > **Important considerations are not included in the main text.**
>
> We thank the reviewer for pointing this out. We have added pointers to the appendix.
>
> > **Classifier architecture is not varied**
>
> We intentionally keep this architecture fixed to isolate the effect of the algorithm itself. In our experiments, we observe a strong empirical correlation between classifier prediction accuracy and the final generation reward, supporting the intuition that classifier quality meaningfully influences controllability.
>
> The architectural component with the greatest impact is the number of discretization buckets. On the compressibility task, using very fine-grained binning (1,000 bins) significantly degraded reward quality (≈ –65.2), presumably because each bin receives too few samples to estimate reliably. In contrast, moderate changes such as 200 vs. 100 bins produced only marginal differences (–30.2 vs. –36.5). Coarser binning (5 bins) yielded performance comparable to our default 50-bin configuration (–16.1). We hypothesize that excessively many bins cause sparse data allocation and unstable bucket-level estimates, while moderate or coarse binning maintains stable signal.
>
> We have added this ablation and discussion to the appendix.
>
> > **SD 1.5 is no longer state of the art.**
>
> Thank you for the suggestion. Our experimental setup follows SVDD \[0\], which also evaluates controllable generation primarily on Stable Diffusion v1.5. We adopt SD v1.5 for two practical reasons: (1) it is substantially smaller (\~ 3.5× smaller than SDXL 3.5), enabling controlled and reproducible ablations of SLCD under limited compute, and (2) it remains one of the most widely used baselines for algorithmic comparisons in controllable diffusion.
>
> Importantly, our goal in this work is to isolate and evaluate the SLCD algorithm itself rather than to benchmark absolute performance across model families. That said, our discrete-domain experiments use the Enformer-based diffusion backbone, which—​to the best of our knowledge—​is state of the art for controlled generation for these tasks. This provides evidence that SLCD generalizes beyond a single architecture class.
>
> We agree that testing on SDXL or SD3 would further strengthen the generality claim, and we view this as a valuable direction for future work.
>
> >  **The scaling parameter for classifier guidance plays a crucial role, but no discussion is provided regarding sensitivity or tuning strategy.**
>
> Thank you for the comment. In our setting, we do not believe that there is a meaningful "tuning strategy" for the scaling parameter $\\eta$. Unlike classifier-guided diffusion methods where $\\eta$ influences both training and inference, SLCD's classifier is trained entirely independently of $\\eta$. Consequently, $\\eta$ affects only the sampling stage and does not alter the learned model.
>
> Because of this decoupling, selecting $\\eta$ does not require traditional hyperparameter tuning. One can simply sample with multiple values of $\\eta$ and directly observe the desired tradeoff between reward improvement and deviation from the base model. Different downstream applications may naturally prefer different points along this tradeoff curve, so there is no single universally "optimal" setting.
>
> To address sensitivity, we have added discussion in the appendix E reporting the numerical range most relevant for our experiments. We observe that values of $\\eta$ between $1$ and $15$ span the regime from minimal intervention to relatively strong guidance.

---

> ### Author Response · Authors · 2025-11-24
>
> #### Response to Questions
>
> > **Does Equation (5) compute KL divergence over the conditional distributions or marginal distributions? If samples are collected under conditional generation in SLCD, is q\_0 treated as a conditional distribution as well?**
>
> Equation 5 computes KL divergence over the marginal distribution at $t=0$.
>
> q\_0 is the distribution generated by the base model, p is the distribution generated by the tuned model. Both are marginal distributions.
>
> > **Could the authors clarify the motivation for using the compression task as an image evaluation setting? What intuition links compression to controllable guidance? Also, could the authors provide more details about the compression task?**
>
> Thank you for the question. We include the compression task because it is a **standard benchmark for controllable diffusion fine-tuning** in prior work \[1,2\]. The task provides a clean setting for evaluating how well an algorithm can optimize a **non-differentiable reward**, which is representative of many real-world controllable generation objectives.
>
> In this task, the reward is defined as the **negative file size** of the generated image. Beyond its practical relevance for storage-constrained applications, compression serves as a challenging testbed: achieving the maximum reward typically requires destroying visual content, and many RL-based approaches indeed converge to such degenerate solutions \[1\].
>
> This makes the task particularly suitable for evaluating controllability. A key feature of SLCD is that, through the guidance strength parameter $\\eta$, we can explicitly balance **reward improvement** against **preservation of the base model’s visual fidelity**. SLCD can reach high-reward regimes while still maintaining recognizable images for moderate $\\eta$, and—when desired—can also match the maximal reward obtained by baselines, though doing so necessarily involves significant distortion.
>
> > **How are hyperparameters for classifier training chosen? What is the computational cost (both sample collection and classifier training)? Does the classifier generalize to unseen prompts?**
>
> The hyperparameters for classifier training were chosen from a light grid sweep over the set of hyperparameters. In particular, we ran a sweep over learning rate, and batch size, selecting the combination which yielded the highest reward for a fixed $\eta$. The computational cost for sample collection is about the same as regular inference, while training is merely a supervised learning problem to predict the correct bucket. This is in contrast to other iterative methods which would require significantly more samples and less informative updates.
>
> We observe that the classifier does a good job generalizing to unseen prompts and point the reviewer to Figure 5 for visual examples for prompts not seen during training.
>
> > **How is the scaling parameter chosen in practice? Is a single value applied across experiments? (such as different rows in Figure 4), or is it tuned per dataset/task?**
>
> Please refer to our reply in weakness 4.
>
> > **In Section 6.3, is FID computed between base SD outputs and SLCD outputs, or between generated outputs and real images?**
>
> The FID is computed between SD outputs and SLCD outputs.
>
> \[0\] X Li, et al. Derivative-Free Guidance in Continuous and Discrete Diffusion Models with Soft Value-Based Decoding
> \[1\] K Black, et al. Training Diffusion Models with Reinforcement Learning
> \[2\] K Clark, et al. Directly Fine-Tuning Diffusion Models on Differentiable Rewards

---

### Official Review · Reviewer_qKFD · 2025-11-03

**Soundness:** 2
**Presentation:** 3
**Contribution:** 2
**Rating:** 4
**Confidence:** 3

**Summary:**

This paper frames controllable generation as KL-regularized reward maximization and introduces SLCD, a fine-tuning-free method that learns an optimal classifier to guide a fixed diffusion model toward the target posterior $p(x)\propto q_0(x)\exp(\omega r(x))$.
SLCD combats the classifier covariate-shift problem by iteratively aggregating on-policy data from its own guided rollouts and trains the guidance via standard supervised learning on a discretized reward distribution.
The theory reduces performance to no-regret online learning: under certain assumptions, the guided sampler’s distribution converges in KL to the optimal solution, with an explicit bound given in Theorem 7.
Empirically, on image aesthetics/compression and DNA/RNA sequence design, SLCD attains higher rewards than training-free baselines at near-base inference cost.

**Strengths:**

1. Under some assumptions, the authors prove that the no-regret learning of $\log R^{prior}$ implies the closedness of the distribution generated by the learned guidance and the target tilted distribution.

**Weaknesses:**

1. The idea of this paper is quite similar to the stochastic optimal control analysis provided in the `adjoint matching` paper:

    A. In `adjoint matching` (actually this is a well-known fact in control literature under the name `Logarithmic Transformations`), it is observed that the optimal value function can be expressed in terms of the uncontrolled prior process. By noting that the classifier $p(y=1 \mid x_t)$ is equivalent to $\exp(-V)$ in `adjoint matching`, one can recover (7) of this paper by using (16) of `adjoint matching` (with the running cost $f\equiv0$).

    B. The classifier $f^n$ in (9) is the same as the optimal control (17) in `adjoint matching`.

    C. Assumption 5 of this paper basically requires the OU process in the forward step of the prior model (pretrained DM) to converge to the equilibrium Gaussian distribution. In this context, the initial random variable $X_0$ and the terminal random variable $X_T$ in under the prior dynamics (pretrained DM) are actually almost independent. This means that the `initial value function bias problem`  in `adjoint matching` is **not** present and there is no need for memoryless noise schedule.

2. The no-regret assumption is rather strong.

**Questions:**

Please see the discussion above.

---

> ### Author Response · Authors · 2025-11-24
>
> We thank the reviewer for the insightful comments. Please refer to our response below:
>
> 1. **Comparison to Adjoint Matching.** We acknowledge the similarity between the optimal classifier and the optimal control. However, **the similarity is to be expected because both papers are solving the same problem, which should have the same optimal solution.** The **techniques** to train the optimal classifier (control) in these two papers **differ significantly**.
>    1. **Different motivations.** Specifically, our paper is inspired by the classifier guidance and DAgger-type reduction to online learning, where we use the fact that the forward and reverse processes of diffusion share the same joint distributions. Please refer to our Appendix A and Lemma 12 for the details. On the other hand, Adjoint Matching starts from a different SOC angle.
>    2. **Distributional approach brings inference-time flexibility**. Instead of training the classifier directly, we train the distribution of the future reward.  (See our equation 8, 9, 11\) This allows us to tune $\eta$ **during inference** to balance between *optimizing the reward* and *staying close to the prior distribution*. In contrast, Adjoint Matching needs to train a different model when choosing a different $\lambda$.
>    3. **Distributional approach allows non-differentiable reward.** Our distributional approach is compatible with both differentiable and non-differentiable rewards, allowing the application to a broader class of problem instances. On the other hand, Adjoint Matching requires a differentiable reward.
>    4. **Light-weight classifier reduce computation consumption.** Instead of tuning the model parameter directly, we train a light weight classifier. This allows us to fine-tune a large model even if the computation resource is limited. Our experiments only require a single A6000 GPU. To fine-tune a 4GB model, we only need to train a 10MB classifier for the aesthetic reward function.
> 2. **Clarification on no-regret learning.** The no-regret assumption is standard in the literature (see e.g. DAgger by Ross et al. 2011 and Foster et al. 2021). Please refer to our line 314 to 319 for the detailed discussion. In the paper, we hope to emphasize our algorithmic design and underlying principles rather than illustrate the no-regret algorithm itself. Importantly, our reduction to online supervised learning enjoys a supervised learning convergence rate, which is much faster than that of the standard RL approach in the literature.

---

### Meta-Review · Area_Chair_5bq7 · 2025-12-30

**Summary:**

The whole DAgger-style on-policy classifier training idea seems to be plausible. However, several reviewers raise concerns about novelty compared with stochastic optimal control. In addition, one reviewer raised concerns about the potentially unfair comparisons with training-free baselines, coupled with missing baselines. Furthermore, the paper lacks computational cost comparisons and ablation studies to justify the efficacy of the proposed method. Thus, I recommend rejection.

**Reviewer Concerns:**

Across the reviews, the main concerns cluster around novelty, evaluation fairness, and missing ablations. Several reviewers feel the paper’s conceptual contribution is not clearly distinguished from existing works. On the experimental side, reviewers worry that the comparisons are not fully fair. The reviewers also asked for detailed training cost comparison results. The authors seem not address these concerns well, thus the paper is not strong enough for acceptance.

**Reviewer Scores:**

Based on the discussion, most reviewers would likely keep their scores unchanged because the core novelty and evidence gaps remain. Overall, the score distribution would still center around borderline-below-accept, supporting rejection.

---

### Decision · Program_Chairs · 2026-01-26

Reject